# Getting funded in a highly fluctuating environment: Shifting from excellence to luck and timing

**Eneli Kindsiko**[1]*, **Kärt Rõigas**[1], **Ülo Niinemets**[2]

**1** School of Economics and Business Administration, University of Tartu, Tartu, Estonia, **2** Estonian Academy of Sciences, Tallinn, Estonia

* eneli.kindsiko@ut.ee

## Abstract

Recent data highlights the presence of luck in research grant allocations, where most vulnerable are early-career researchers. The national research funding contributes typically the greatest share of total research funding in a given country, fulfilling simultaneously the roles of promoting excellence in science, and most importantly, development of the careers of young generation of scientists. Yet, there is limited supply of studies that have investigated how do early-career researchers stand compared to advanced-career level researchers in case of a national research grant system. We analyzed the Estonian national highly competitive research grant funding across different fields of research for a ten-year-period between 2013–2022, including all the awarded grants for this period (845 grants, 658 individual principal investigators, PI). The analysis was conducted separately for early-career and advanced-career researchers. We aimed to investigate how the age, scientific productivity and the previous grant success of the PI vary across a national research system, by comparing early- and advanced-career researchers. The annual grant success rates varied between 14% and 28%, and within the discipline the success rate fluctuated across years even between 0–67%. The year-to-year fluctuations in grant success were stronger for early-career researchers. The study highlights how the seniority does not automatically deliver better research performance, at some fields, younger PIs outperform older cohorts. Also, as the size of the available annual grants fluctuates remarkably, early-career researchers are most vulnerable as they can apply for the starting grant only within a limited "time window".

## Introduction

Estonian research system has been described as a success story of a nationally funded research system [1]: being "one of the world's tiniest developed nations" with only 1.3 million inhabitants, "its research community comprises just over 3,000 full-time researchers in academia". The country relies heavily on European Union (EU) structural funds as common for post eastern-bloc countries in EU who "used generous EU funds to modernize their science infrastructures", and are now facing great challenges if EU support decreases [1].

**Data Availability Statement:** All relevant data are within the paper and its Supporting Information files.

**Funding:** 1) Dr Kärt Rõigas: Funder: Horizon 2020 Framework Programme Grant 822781, "Growth

Welfare Innovation Productivity" 2) Dr Eneli Kindsiko: Funder: Estonian Research Council Grant PRG791, "Innovation Complementarities and Productivity Growth" The funders did not play any role in the study design, data collection and analysis, decision to publish, or preparation of the manuscript.

**Competing interests:** The authors have declared that no competing interests exist

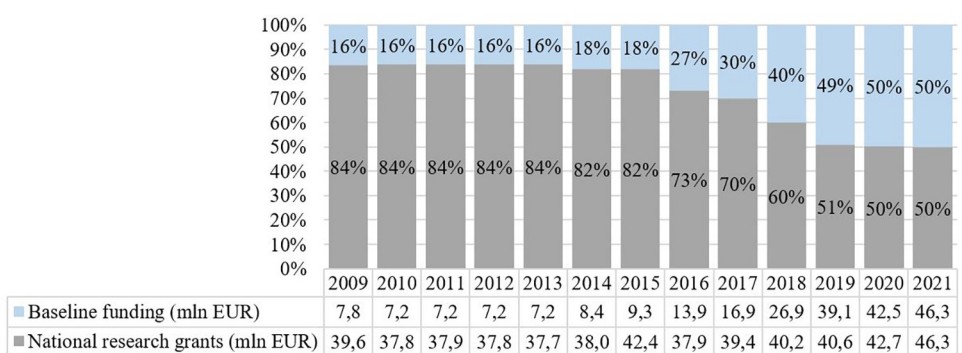

**Fig 1. Proportions between baseline funding and national research grants between 2009–2021 [4].**

Despite seen as a success story, the overall Estonian state funding in research, development and innovation (RDI) has been moderate [2], on average, 0.66% of gross domestic product (GDP) for the period 2013–2020 (declining from 0.81% in 2013 to 0.46% in 2016 [3]. Estonian research has been highly dependent on grant-based competitive funding for individuals and research teams; the baseline state funding for research institutions has been low (16% in 2009) to moderate (50% in 2021). The baseline funding for institutions is also competitive and is based on the number of publications and grant funding obtained by researchers of the given institution and number of PhD theses defended. The rise of the share of the baseline funding has been a science policy decision to reduce the job insecurity in research due to too high reliance on grant funding. However, a large part of baseline funding is currently used to support the infrastructure and daily management of institutions, not for remuneration (Fig 1).

Moderate state funding, and stagnating funding for multiple years in sequence, has made it increasingly difficult for early-career researchers to obtain a national research grant and start an independent research career [2]. Alberts et al. [5] have stated how in the case of low success rate of funding, scientists spend disproportional amount of time on writing (non)successful grants, leaving much less time for actual research. Furthermore, it has been reported how funding rates below 20% are likely to drive at least half of the active researchers away from nationally funded research [6]. In Estonia, the average success rates of obtaining a national research grant is around 20%, and in some fields, it has been even as low as 6% for early-career researchers in 2020 [7]. A great share of national research systems allocate grants as financial incentives as to increase the research performance of its researchers [8, 9], yet for researchers in Estonia, national research grants often constitute the only source for salary of the team. In a highly competitive funding environment, it is important that the concept of excellence is rigorously applied in allocating the limited funding. However, due to the impact of year-to-year fluctuations in how much money will be available for new grants, the success becomes partly uncoupled from excellence as the level of competition varies among years and the success will be largely driven by being at a right place at a right time (RPRT). RPRT is expected to have the strongest impact on early-career researchers, whose continuation in the research system often hinges on obtaining the national grant funding.

In this study, we claim that these competitive national grant systems are most fatal to the early-career researchers as they are most vulnerable to any kind of policy changes and fluctuations in annual R&D money. Based on the full list of Estonian national grant applicants between 2013–2022, we compare the success in obtaining the grant funding among early-career researchers compared to advanced-career researchers across different fields of science. We define early-career and advanced-career researcher according to the eligibility criteria of

the grant funding schema, e.g. one can apply for early-career grant between two to seven years after obtaining a PhD and there is no age limit in doing so. Having a complete list of a ten-year period of Estonian research grant funding and biographic information of all principal investigators (PIs) constitutes a unique resource that allows us to obtain a holistic picture of the operation of a national grant system and draw conclusions of wider importance.

Multiple studies have signposted that in national research grant evaluation systems certain groups of applicants often have an advantage in attracting research funding [10–12]. Non-objective unfair advantages blur the connection between research excellence and funding success, and can have especially devastating consequences under extreme level of competition for funding. Most discussed characteristics of these more advantageous groups are the seniority or the age of the principal investigator, the previous research success, and also the past track record in attracting research funding. Seniority is often seen as the proof of research excellence; researchers at advanced career levels often have a larger number of national and international collaborators, resulting in higher number of publications and greater citation counts [9, 13, 14]. However, this is not always the case especially when the metrics are standardized with respect to the time spent doing the research. Furthermore, there is evidence that all other metrics being equal, more senior applicants gain on average higher scores in grant evaluation process [14, 15], despite that the younger generation might even surpass the older generation in terms of research productivity [16].

Second, highly connected to the previous factor is the number of publications, which is expected to grow with seniority [10, 17]. One of the few studies that has explored the question has focused on the applicants of ERC Starting Grant [18]. The study revealed how in case of early-career researchers, the average annual number of publications that the applicants reported was around 3–5 [18]. To be eligible for the ERC Starting Grant the principal investigator should have "2–7 years of experience since completion of PhD" [19], which means that early-career scholars applying for ERC Starting Grant ought to report very good track of research excellence prior to their first funding. That said, even as an early-career researcher, in order to obtain the grant, you already have to be an excellent researcher [20].

Third, also connected to the first and the second factor, is the previous success needed in attracting research money–money follows money [10, 21]. Early-career researchers do not usually benefit from what is the advantage of senior researchers—Cumulative Advantage effect [21], where previously successful researchers in attracting grants face better odds in gaining a new grant compared to the ones, who have not enjoyed research funding [6, 10]. Yet, current stock of scholarly knowledge has not revealed whether such cumulative advantage is present also among early-career researchers. In the Netherlands, Bol et al. [21], analyzing the scores of research proposals by early-career researchers stated that "candidates who won prior awards are evaluated more positively than non-winners" [21], but it is unclear how general is this suggestion. Thus we ask, have the early-career level researchers expected to benefit from previous success in attracting research money?

Considering all points above, we pose the following research questions:

RQ1: Does the average age at the time of the grant differ by grant level (early vs. advanced)?

RQ2: Does the average scientific productivity differ by grant level?

RQ3: Does the previous success at gaining a grant differ by grant level?

The remainder of the paper proceeds as follows: next section presents an overview of the Estonian national grant assessment; explains the data and methodology; the empirical results and interpretation with discussion are followed next; lastly, key takeaways from our findings are reported under conclusions, together with the limitations and some future research directions.

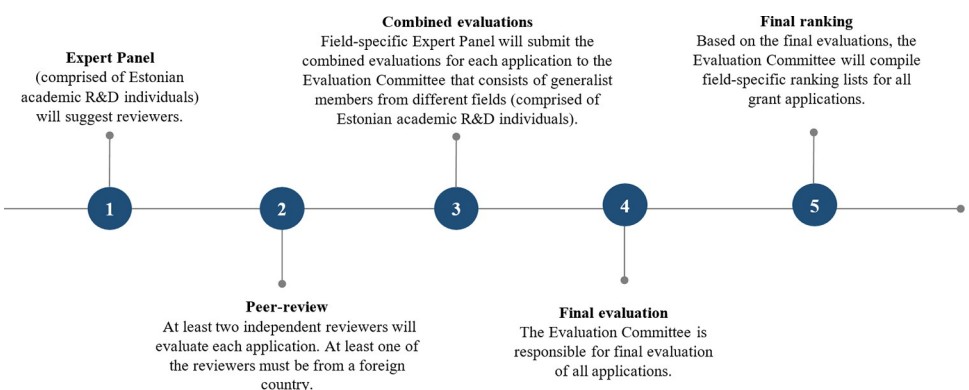

**Fig 2. Overview of the evaluation process of national research grants in Estonia [22].**

## Materials and methods

### Decision process within the Estonian Research Council

The decision process within the Estonian Research Council starts with a grant application from a researcher via Estonian Research Information System once a year. After the submission, the evaluation process proceeds according to the steps illustrated on Fig 2.

The threefold evaluation procedure (ad hoc reviewers, field-specific national Expert Panels, and national Evaluation Committee) is supposed to even out the reviewer-specific bias, but the individual national members of the expert panels only assess the proposals assigned to them rather than all proposals submitted to the given panel (differently, for example, from the practice in European Research Council grant assessment). Therefore, the rankings are primarily based on averages of the scores provided by the Reviewers, and the panel consensus report does not consider expertize of all the panel members. This includes a large random component in the final scores.

### Database construction

We constructed a database including all competitive research projects funded by the Estonian Research Council (www.etag.ee) between 2013–2022. During this period, altogether 845 research grants from all fields of science have been funded. The projects were broadly divided among the two major categories, early-career and advanced-career grant, based on the research experience level of the PI (Table 1). In addition, the funding scheme of the projects somewhat differed among the years (Table 1). In particular, early-career grants base on the "money follows the researcher" logic, while most of the advanced-career level grants (except the discovery grant) do not. These rules were in place to assure the continuity of the research topic in the given research institution. Thus, in addition to research excellence, the funding decisions also considered the science policy decision.

The database included the following fields: project type (advanced vs. early, funding scheme; Table 1), project title, year of funding, duration of the project, name and birthdate of the PI, date of PhD defense of the PI, and the number of peer-reviewed articles published by the PI by the time s(he) applied to the grant. The birthdates and dates of PhD defenses of PIs were obtained from public sources, including the Estonian Research Information System (www.etis.ee). The number of peer-reviewed articles published by the PI were obtained from the Estonian Research Information System (www.etis.ee), which is detailed academic CV or academic database of all academics working in Estonia. Estonian Research Information System is also the place, where applicants submit their application for the grants analyzed in this

**Table 1. Description of Estonian national grant types.**

| Grant level | Grant type | Eligibility | Funding period | Application years (ongoing or has ended) | Average sum EUR for a year |
|---|---|---|---|---|---|
| Early-career level | Start-up grant | Doctoral degree obtained 2–7 years prior to the 1st of January of the year the project is scheduled to begin. The person has to be employed full-time at the Estonian academic institution, and have a place of work in Estonia at the time of implementing the project. | Up to 4 years | Since 2012-... | 62 989 |
| | Postdoctoral research grant | Doctoral degree obtained from an Estonian university no more than five years prior to the 1st of January of the year of the call. Scholars of any nationality are eligible. Only outgoing fellowships to gain postdoctoral experience at a foreign R&D institution are awarded. | 12–36 months | Since 2013-... | 37 446 |
| Advanced–career level | Institutional research grant | Grants provided for researchers to continue their research career, ensuring high-quality research and support research teams. No limitations in terms of applicant age, nor the time between the call and the year doctoral degree obtained. The principal investigator has to be employed full-time at the institution and have a place of work in Estonia at the time of implementing the project. | Up to 6 years | 2012–2015 | 160 930 |
| | Discovery grant | | Up to 4 years | 2012–2016 | 55 706 |
| | Team grant | | Up to 5 years | Since 2017-... | 215 894 |

paper. That said, the application will base on CV (including the articles) that published in Estonian Research Information System.

We use full publication counts instead of counts weighted by the author number as it is difficult to state what was the actual contribution of each author. The problem has been well-addressed in literature as it poses severe difficulties for the scientometric analysis [23, 24]. Also, the reviewers of the grant applications have the same data–full counts of articles per applicant. We also considered taking into an account the ranking of authors, who is the first, second, etc., yet this is where disciplinary differences come in [25]. In lab-based disciplines it is common to have a long list of authors, whereas the first author is often an early-career researcher, a PhD student or a postdoc, who has carried out most of the experimental work, and the last place in the author list is typically reserved for the PI of the team, although the position of authors can vary depending on how authors weight planning, experimental work, data analysis, writing and supervisory monitoring of the whole experiment. Yet, in humanities and in social sciences single-author publications are still most widespread [26, 27].

## Measures and derived characteristics

For each PI, we calculated the age at the time of grant, $A_g$, and the time from obtaining the PhD. In addition to the total number of peer-reviewed articles, $n_t$, we calculated a normalized estimate, $N_n$ ($yr^{-1}$), that corrected the number of articles with the age of the PI using the following equation:

$$N_n = n_t/(A_g-25),\tag{1}$$

where the constant 25 yr considers the circumstance that most researchers in Estonia have their first scientific paper published at the time of the PhD, typically at the age of 24–26 yr. The normalized value, $N_n$, accounts for the cumulative effect of seniority due to which the young grant holders typically have a lower $n_t$ than older PIs, except under exceptional circumstances.

We used the statistical software package Stata 14.0 (StataCorp LLC, College Station, Texas, USA) to analyze the data. Two sample independent samples t-test was used to compare the average values (Tables 3–5) of early and advanced-career level, the differences between the average levels across research fields were tested with two-sample Wilcoxon rank-sum test due to the limited number of observations in each of the research fields. For the correlation analysis, Pearson correlation coefficient was used.

**Table 2. Information of the number of grants, distribution of gender and fields of research across grant types (early/advanced) for grants awarded by the Estonian Research Council between 2013–2022.**

| | Early-career | | | Advanced-career | | |
|---|---|---|---|---|---|---|
| | Mean±SE | Median | CV | Mean±SE | Median | CV |
| Average number of grants per year | 33.0±0.6 | 33 | 33.0 | 51.5±1.0 | 54 | 42.4 |
| Average number of grants per year by field of research | | | | | | |
| *Natural Sciences* | 16.2±0.6 | 16.5 | 47.1 | 23.6±0.8 | 25 | 48.9 |
| *Engineering and technology* | 4.3±0.3 | 4.5 | 42.5 | 5.0±0.5 | 4 | 74.2 |
| *Medical and health sciences* | 4.2±0.3 | 4 | 51.2 | 9.6±0.4 | 9.5 | 42.6 |
| *Agricultural and veterinary sciences* | 1.4±0.2 | 1 | 37.4 | 3.3±0.4 | 3.5 | 49.0 |
| *Social Sciences* | 3.8±0.3 | 4 | 47.7 | 4.4±0.3 | 4 | 43.1 |
| *Humanities and the arts* | 3.8±0.3 | 3 | 45.4 | 6.9±0.4 | 7 | 47.6 |

Note: SE–standard error; CV–coefficient of variation. The fields of science are based on the Frascati classification [28].

## Results

### Estonian science grants between 2013–2022: General patterns

Out of the total number of 845 grants, 330 were given at the early and 515 at advanced-career level (Table 2). Compared to other research fields, the field Natural sciences is by far the biggest in terms of grant holders and the field Agricultural and veterinary sciences is the smallest, both at early and advanced-career level (Table 2).

The Estonian Research Council has a fixed annual budget that has to be used up in a given calendar year. Given that the research grants by Estonian Research Council are provided for a 4–5 year period, that said, the annual budget always has a pre-booked sum, which may not be used for new grantees, but are meant for the ongoing projects. In Estonia, during the analysis period of 2013–2022, the annual budget that was available for new grants fluctuated almost 7-fold from 2.74 mln EUR in 2016 to 18.3 mln EUR in 2019 (Fig 3).

There was a change in research policy–between 2016 and 2018, Estonian Research Council closed several smaller advanced level grant types (discovery and institutional grant), and increased remarkably the maximum funding amount of the remaining grants–this explains

**Table 3. Average age of PI at the time of the grant across different fields of science for grants awarded by the Estonian Research Council between 2013–2022 (Research question 1).**

| | Early-career | | | Advanced-career | | |
|---|---|---|---|---|---|---|
| | Mean±SE | Median | CV | Mean±SE | Median | CV |
| Average age at the time of gaining the grant (yr) | 35.5±0.3[a] | 35.1 | 12.7 | 50.9±0.4[b] | 50.2 | 18.1 |
| Average age at the time of gaining the grant by field of research (yr) | | | | | | |
| *Natural Sciences* | 34.4±0.3[a, A, B] | 34.3 | 11.2 | 50.7±0.7[b] | 50.2 | 19.6 |
| *Engineering and technology* | 35.0±0.6[a, A, B] | 34.6 | 11.8 | 49.2±1.4[b] | 48.1 | 20.3 |
| *Medical and health sciences* | 35.1±0.5[a, A, B] | 34.6 | 9.1 | 52.4±0.8[b,A] | 51.6 | 15.8 |
| *Agricultural and veterinary sciences* | 36.8±1.3[a] | 36.7 | 9.7 | 53.6±1.9[b,A] | 54.8 | 15.6 |
| *Social Sciences* | 38.8±0.8[a] | 39.2 | 13.3 | 48.7±1.3[b] | 46.4 | 17.9 |
| *Humanities and the arts* | 37.9±1.1[a] | 36.1 | 15.9 | 51.2±0.9[b,A] | 50.5 | 14.9 |

Note: Means with different lowercase letters are significantly different among early and advanced levels (based on the sample size, two independent samples t-test or two-sample Wilcoxon rank-sum test was used, $P < 0.05$); Means with different uppercase letters are significantly different among fields of science within the given grant level A–differs significantly from the average of Social Sciences, B–differs significantly from the average of Humanities and the arts (comparison between the research fields is within the particular career level, two-sample Wilcoxon rank-sum test, $P < 0.05$).

**Table 4. Average number of peer-reviewed articles and age-normalized number of articles published by grant holders for different grant types across different fields of science for grants awarded by the Estonian Research Council between 2013–2022 (Research question 2).**

| | Early-career | | | Advanced-career | | |
|---|---|---|---|---|---|---|
| | Mean±SE | Median | CV | Mean±SE | Median | CV |
| Average number of articles ($n_t$) | 13.5±1.0[a] | 10 | 131.9 | 55.4±3.2 | 36 | 131.3 |
| Average normalized number of articles ($N_n$, yr$^{-1}$) | 1.45±0.12[a] | 1.05 | 142.5 | 2.36±0.18 | 1.47 | 169.2 |
| Average number of articles by field of research | | | | | | |
| *Natural Sciences* | 16.0±1.8[aA] | 11.0 | 145.4 | 70.2±6.0[A, C] | 48.0 | 131.4 |
| *Engineering and technology* | 13.5±1.6[aA] | 12.0 | 79.5 | 51.7±8.0[A, D] | 29.0 | 109.3 |
| *Medical and health sciences* | 14.2±1.5[aA] | 11.0 | 69.2 | 66.0±5.1[A] | 52.0 | 76.2 |
| *Agricultural and veterinary sciences* | 16.9±2.1[aA] | 16.5 | 38.8 | 52.9±11.7[A] | 39.5 | 99.3 |
| *Social Sciences* | 8.4±1.4[a] | 7.0 | 100.9 | 20.9±3.3[B] | 15.0 | 103.9 |
| *Humanities and the arts* | 5.4±0.7[a] | 4.0 | 75.3 | 15.5±2.2 | 10.0 | 118.3 |
| Average normalized number of articles by field of research | | | | | | |
| *Natural Sciences* | 1.80±0.22[aA] | 1.28 | 151.1 | 3.24±0.37[A] | 1.79 | 172.2 |
| *Engineering and technology* | 1.55±0.20[A] | 1.11 | 81.9 | 2.09±0.24[A] | 1.44 | 80.0 |
| *Medical and health sciences* | 1.45±0.14[a, A] | 1.27 | 61.9 | 2.39±0.15[b,A] | 2.10 | 62.7 |
| *Agricultural and veterinary sciences* | 1.75±0.43[a,A] | 1.43 | 70.1 | 1.85±0.32[a,A] | 1.48 | 75.2 |
| *Social Sciences* | 0.66±0.11[a] | 0.45 | 98.5 | 0.89±0.11[B] | 0.78 | 81.3 |
| *Humanities and the arts* | 0.46±0.05[a] | 0.42 | 58.2 | 0.65±0.10[b] | 0.46 | 126.3 |

Note

a–differs significantly from the average of advanced level (based on the sample size, two independent samples t-test or two-sample Wilcoxon rank-sum test was used, $P < 0.05$)

A–differs significantly from the average of Social Sciences and Humanities and the arts

B–differs significantly from the average of Humanities and the arts

C–differs significantly form the average of Engineering and technology

D–differs significantly from the average of Medical and health sciences; (comparison between the research fields is within the particular career level, two-sample Wilcoxon rank-sum test, $P < 0.05$).

**Table 5. The share of PIs who have gained the grant several times across different fields of science for grants awarded by the Estonian Research Council between 2013–2022 (Research question 3).**

| | Early-career | Advanced-career |
|---|---|---|
| | Mean±SE | Mean±SE |
| The share of PIs who have gained the grant several times | 25.5±2.4[a] | 56.1±2.2 |
| The share of PIs who have gained the grant several times by field of research | | |
| *Natural Sciences* | 23.5±3.3[a] | 58.1±3.2 |
| *Engineering and technology* | 20.9±6.3[a] | 54.0±7.1 |
| *Medical and health sciences* | 35.7±7.5[a] | 59.4±5.0 |
| *Agricultural and veterinary sciences* | 40.0±16.3 | 70.0±10.5 |
| *Social Sciences* | 31.6±7.6 | 47.7±7.6 |
| *Humanities and the arts* | 17.6±6.6[a] | 47.8±6.1 |

Note

a–differs significantly from the average of advanced level (based on the sample size, two independent samples t-test or two-sample Wilcoxon rank-sum test was used, $P < 0.05$).

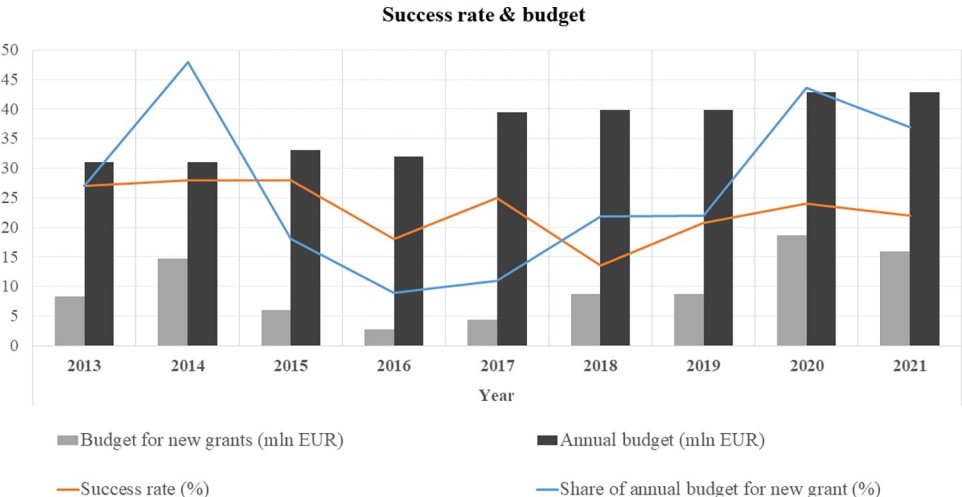

**Fig 3. The importance of timing in applying for the grant: Success rate and annual budget available for new grants.** Percentage values shows the success rate in the given year.

the jump in annual grant sum from 2018. To bring an example, the max sum of team grant today is 257 125 EUR per year (average is 212 117 EUR), which makes a total of 1 285 625 EUR for a 5 year grant period. An increase in average grant sum explains also a decrease in success rates as a smaller number of researchers gain grants, though the total funding is greater.

The success rate of the grants awarded by the Estonian Research Council is low, the average success rate for the early level is 24.5% and for advanced level 21.4%. At the same time, the success rate is highly fluctuating over time (Fig 4). The average annual size of the grants has increased both for early and advanced-career level after the change of the grant policy. It can be seen from Fig 4 that the average annual size of the grants had a decreasing trend at the advanced-career level before the change in grant policy.

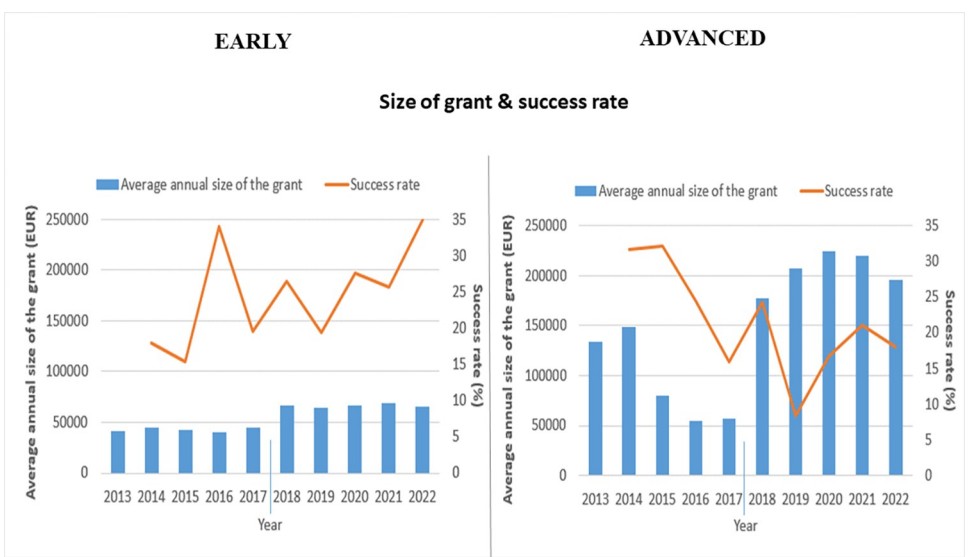

**Fig 4. Yearly size of the grant and success rate for early-stage and advanced-stage grant types across all grants awarded by the Estonian Research Council between 2013–2022.** The line between 2017 and 2018 indicates the change in grant policy.

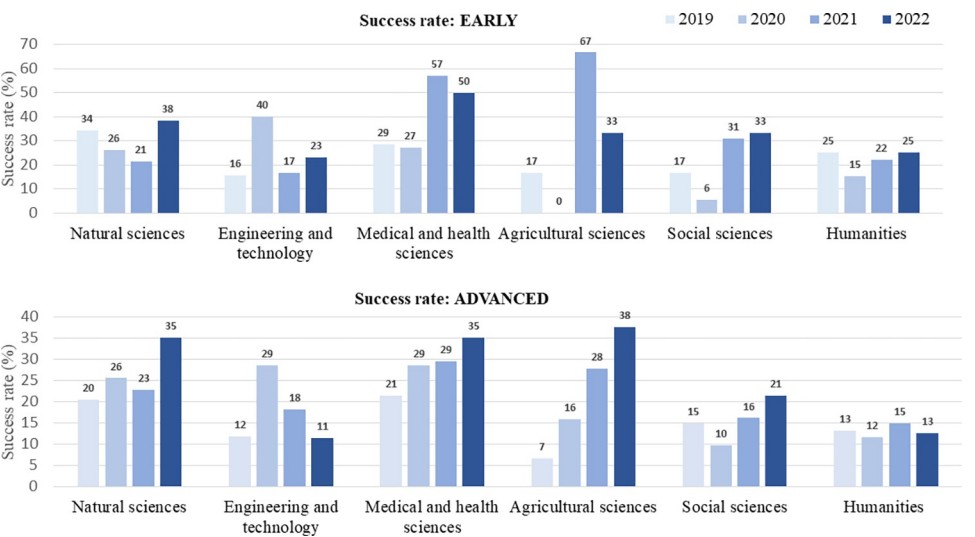

**Fig 5. Annual success rates across fields and grant levels.**

Even more heterogeneity is witnessed when we zoom into the field level. Fig 5 reveals how fluctuations in success rate are remarkable and on average, humanities and social sciences have lower success rates both at the early and advanced level. Most striking is the 6% success rate of early-career applicants in social sciences in 2020. Across all years and fields, fluctuations in success rates can be more than 60%. Most strikingly, the success rate has varied between 0–67% for early-career level grants in Agricultural sciences. In fact, the strongest year-to-year fluctuations have occurred at the early-career level: both within the same discipline, and across disciplines (Fig 5).

## Age of the successful grant applicants

The average age of the PI at the time of the grant was 15.4 years lower for early than for advanced level grants (Table 3 for average age of different grant types), although there were relatively old PIs for early-career level, and young researchers for advanced-career level. Indeed, across all grants, the age of the grant holders has varied from 26 to 76 yr. at the time of awarding the grant, and from 26 to 61 yr. at early level and from 29 to 76 yr. at advanced level (Table 3). For all the fields of research and also for the whole sample, the average age was higher for advanced-career level (Table 3). The average age was the lowest in Social sciences and differed significantly from the average of Medical and health sciences, Agricultural and veterinary sciences and from the Humanities and the arts at early-career level (Table 3). On the early-career level, the average age was the highest in Social sciences and the lowest in Natural sciences.

The shape of the distribution of age of the PI at the time of the grant differed between early and advanced-career levels (Fig 6). While the distribution of the age was close to normal at the advanced level, it was positively skewed and had a much greater kurtosis at the early-career level (Fig 6).

For the early-career grants, the average age of PIs significantly differed among Frascati fields (two-sample Wilcoxon rank-sum test; Table 3, S1 Fig). In particular, the average age of PIs for Natural sciences was smaller than that for Social sciences and from Humanities and the arts (Table 3). The average age of PI for Social sciences was greater than that Engineering and technology and Medical and health sciences (Table 3). In the case of advanced-career grants,

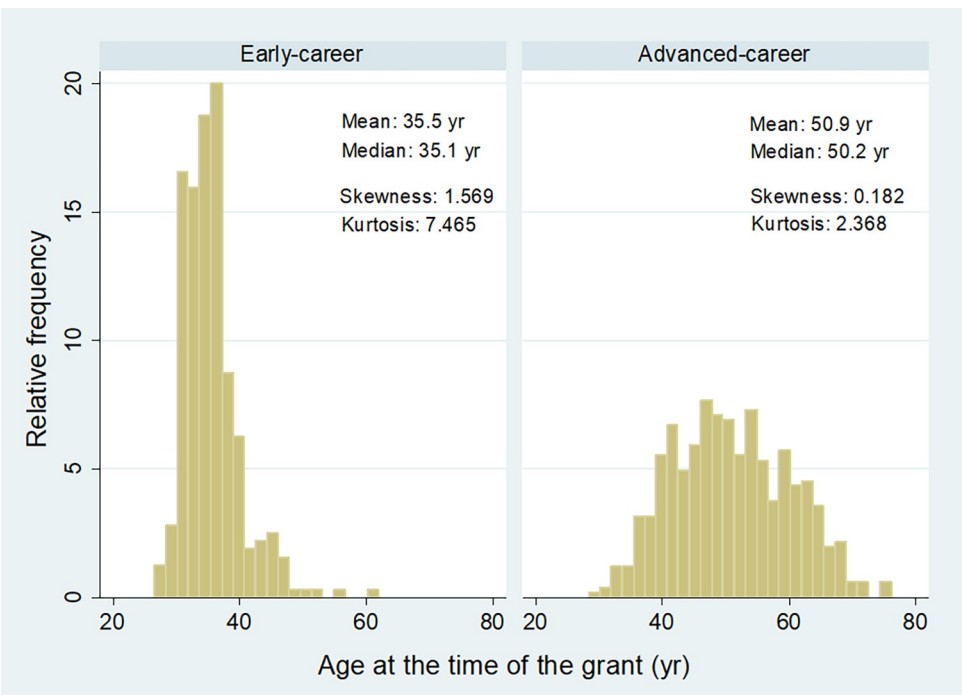

**Fig 6. Distribution of the age of the PI at the time of the grant and distribution statistics for early and advanced grant types across all grants awarded by the Estonian Research Council between 2013–2022.**

the average age for Social sciences was significantly lower compared to Medical and health sciences, Agricultural and veterinary sciences and from the Humanities and the arts (Table 3).

## Research productivity of PIs

As explained in Materials and methods (derived characteristics part), to get a more comparable value of the number of articles for different age groups, we normalized the number of articles with the age of the grant holder (Eq. 1). After normalization, the distribution was positively skewed for both the early and advanced levels (Fig 7), indicating a large share of researchers with limited productivity for both career levels. After logarithmic transformation of normalized number of articles, the distributions are close to normal distribution (smaller pictures on Fig 7). The distribution of the original data of the number of the articles with its logarithmic transformation is shown in S2 Fig.

In the case of non-normalized estimates, the average number of articles was more than four-fold lower for PIs at early-career level than for advanced-career level (Table 4). In the case of normalized articles, the difference was less, it was around 1.6 times higher at advanced-career level (Table 4). At both career levels, the number of articles had a very high variation indicated by the coefficient of variation (Table 4). The high level of variation partly reflected differences in publication practices among the research fields. Both at early and advanced-career level, Social sciences and Humanities and the arts had the lowest average number of articles, this pattern also held for the average number of normalized articles (Table 4). The highest average number of normalized articles was in Natural sciences (both at early and advanced-career level), but the variation was also the highest in this field (Table 4).

The temporal trends of the average number of normalized peer-reviewed (1.1) articles were different for the early and advanced-career levels. At the early-career level, the average number

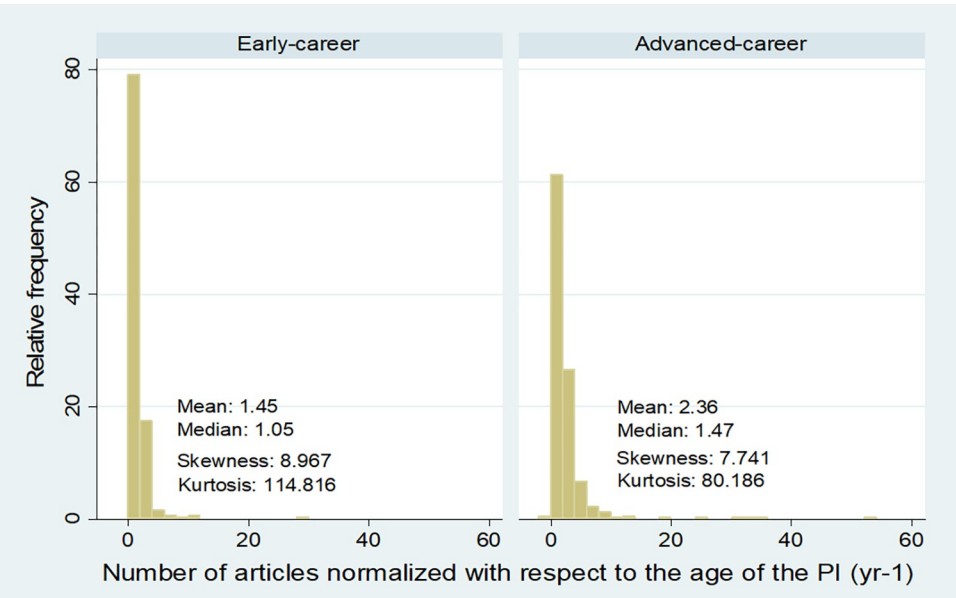

**Fig 7. The distribution of the normalized number of peer-reviewed articles (1.1 articles) and distribution characteristics for different grant types through all grants awarded by the Estonian Research Council between 2013–2022.** The article number was normalized with respect to the age of the PI (Eq. 1). The insets show the distributions after logarithmic transformation.

of articles was stable over time and the averages did not differ across the years (Fig 8). In contrast, at advanced grant level, the average number of normalized peer-review (1.1) articles was significantly higher in years 2019, 2020 and 2021, compared to the period of 2013–2017 and the year 2022.

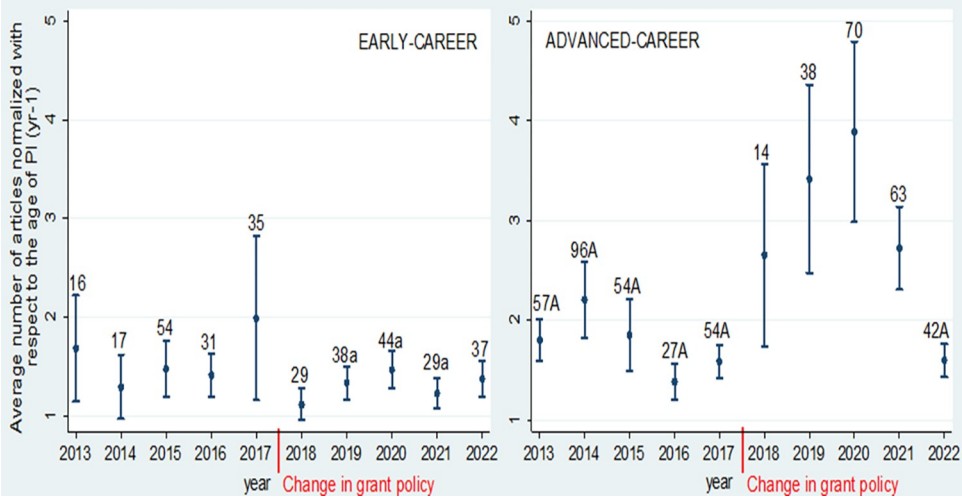

**Fig 8. The average ± SE number of peer-reviewed articles normalized with respect to the age of the researcher (Eq. 1) for PIs of early and advanced-career grants.** The number of grants gained in particular year is given above the intervals. a–differs significantly from the average of advanced level; A–differs significantly from the average of years 2019, 2020 and 2021 (comparison between the years is within the particular career level, two-sample Wilcoxon rank-sum test, $P < 0.05$). EP-Expert Panel; EC-Evaluation Committee, both play important role in final decision making over funding allocation.

There was a change in the grant policy in 2018. At the early level, in 2017, the average number of the articles was the highest and the variation of the grant holders in terms of the number of articles was also the highest (it can be seen from the intervals on Fig 8). After the change in the grant policy, the number of grants given at the advanced level dropped for that particular year. At the same time, the level of performance of the grant holders started to show a growing trend at the advanced level until year 2020, both the number of grants given and the level of performance started to decrease after 2020. We added another layer to the Fig 8 to give possible interpretation of the mentioned fluctuation of the performance–namely, both the Expert Panel and Evaluation Committee change after a few years. As both the reviewer selection and final decision over funded grants is done by these two evaluation bodies (comprising of solely Estonian national R&D experts), we see this as a significant possibility for subjective element in grant evaluation. In a country with only few thousand of academic employees, having evaluation experts from the same pool of scientists is bound to trigger conflicts of interests. Overall, this is also one of the possible contributing factors within the RPRT effect to happen–in addition to applying at the year when more money is available, also the composition of the evaluation committee may have an effect. To our knowledge it is not very common to have evaluation bodies from the same system, even from the same academic units.

The non-normalized number of articles was positively correlated with the age of the grant holder after removing the outliers ($r = 0.513$, $P < 0.001$). Normalization of article number by age of the PI (Eq. 1) essentially removed the statistical difference between the average number of articles for early and advanced level grant holders in the fields of Engineering and technology, Agricultural and veterinary sciences and in Humanities and the arts (Table 4, S3 Fig).

The outliers in the normalized article number vs. age relationship all belonged to the scientific fields characterized by high frequency of publishing of multi-author papers in Natural Sciences. The highest values of normalized number of articles belong to grant holders not from universities, but from governmentally financed research institutions, e.g. National Institute of Chemical Physics and Biophysics (Fig 9).

What is striking is the erosion of the term early-career researcher among some fields of science (see Fig 9, middle figure). In Humanities and Social sciences, 24% of early-career grants have been given to PIs older than 40. Furthermore, one PI from Natural sciences gained early-career grant at the age of 61 and during application year had already 48 peer-reviewed publications. From one side, age limits have not been added to grant schemas as to lessen the punishing effects to people with career breaks (e.g. maternity leaves), from the other side, the system benefits those who have been decades with the academic system (active in publishing), yet defended their doctoral degree very late.

Also, at the advanced career-level, 16% of PIs are over 60 yr., 9% are beyond retirement age and 1% over 70yr., which means that these advanced level grants are working not as career development, but as the end-of-career grants. As Estonian universities do not have obligatory, but recommended retirement age (65yr.), this may be also behind the presence of older PIs. The official retirement age in Estonia is 64yr.

The share of the PIs with the lowest values of the normalized number of articles, below 0.5, was the highest among Social sciences and Humanities and the arts (Table 4). In these fields, the average article numbers were also lower compared to other fields (Fig 10). In addition, the average non-normalized number of articles was also lower for Social sciences and Humanities and the arts compared to the other fields (Table 4).

The number of normalized articles for Social Sciences and the Humanities and arts remained similar for all age groups (Fig 10). In contrast, in Engineering and technology and Medical and health sciences a growing trend was evident with the average number of normalized articles increasing in older age groups compared to the younger ones (Fig 10). Natural

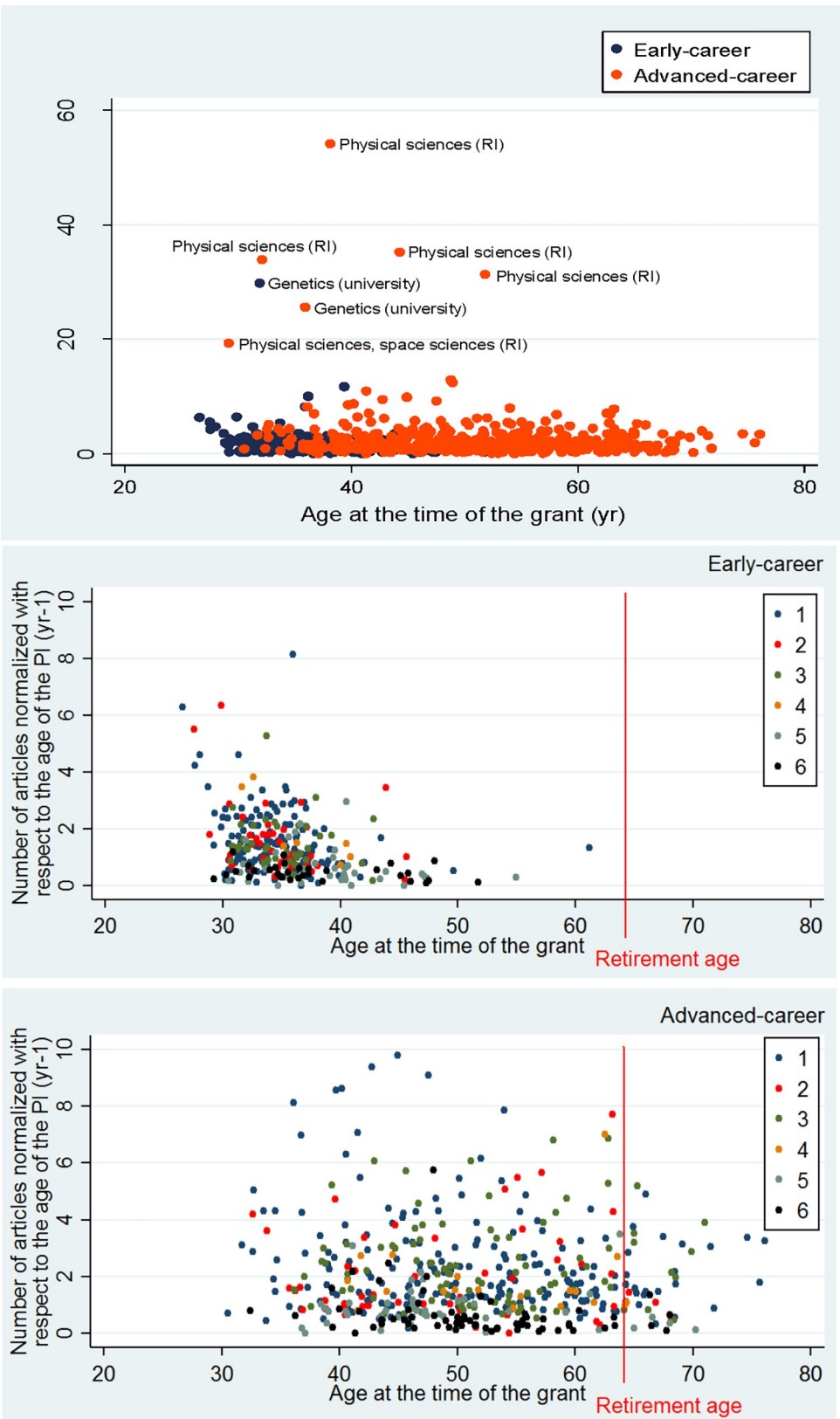

**Fig 9. The relationship between the normalized number of peer-reviewed articles and the age of PI at the time of the grant for different grant types across all grants awarded by the Estonian Research Council between 2013–**

**2022.** Peer-reviewed articles as in Fig 7. The article number was normalized with respect to the age of the researcher by Eq. 1. RI–research institution. Above–all grants with outliers; middle–early-career grants, outliers left out below–advanced career grants, outliers left out.

sciences stand out with a different trend as in younger age groups, the average number of normalized articles was higher compared to the age group of 61 and older (Fig 10). Thus, in Natural sciences, those gaining the grant at the age of 61 or older, had on average lower performance than younger grant holders in the same field.

## Determinants of the success of recurrent funding

Among the PIs, the frequency of grant holders who have received the grant multiple times was about two times greater for advanced than for early-career level (Table 5). The share of researchers who have obtained multiple grants at the early-career level is higher for Agricultural and veterinary sciences, Medical and health sciences and Social Sciences (Table 5). At the advanced-career level, the share is the lowest in Social Sciences and in Humanities and the arts (Table 5).

The difference in gaining grants several times between early and advanced-career level holds for most of the research fields with one exception–the Social Sciences that is the only field of research, where the share of PIs with several grants is similar for early and advanced-career level (Fig 11).

Neither within early-career level nor within advanced-career level, there are no significant differences in average age at the time of gaining PhD in relation to success of obtaining the grant once or several times between the fields of research (Fig 12).

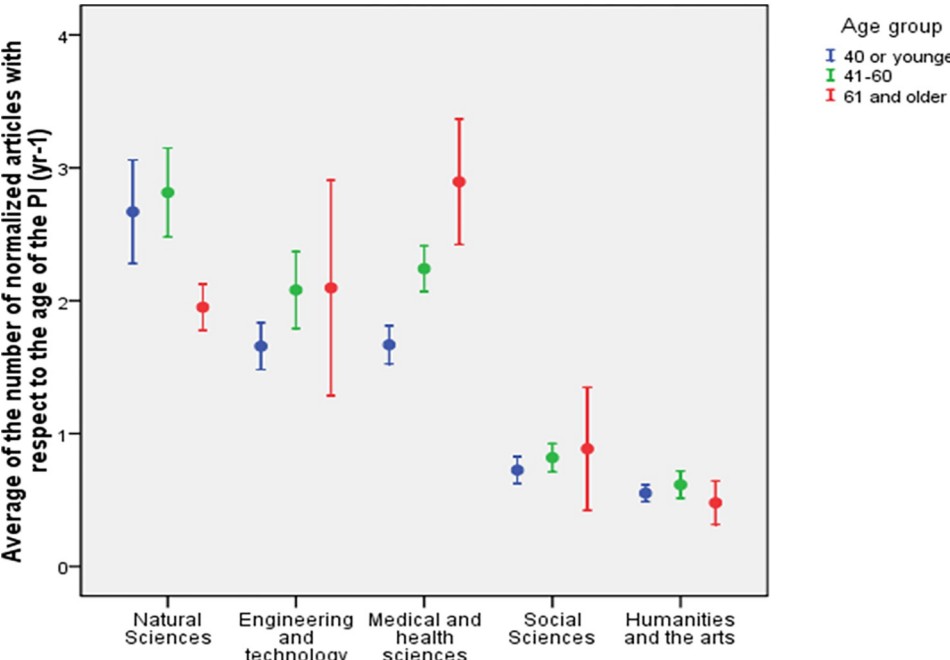

**Fig 10. The average (± SE) number or peer-reviewed articles normalized with respect to the age of the PI (Eq. 1) across age groups and Frascati fields through all grants awarded by the Estonian Research Council between 2013–2022.** Agricultural and veterinary sciences are not shown because of low number of grant holders in the oldest age group (5 PIs).

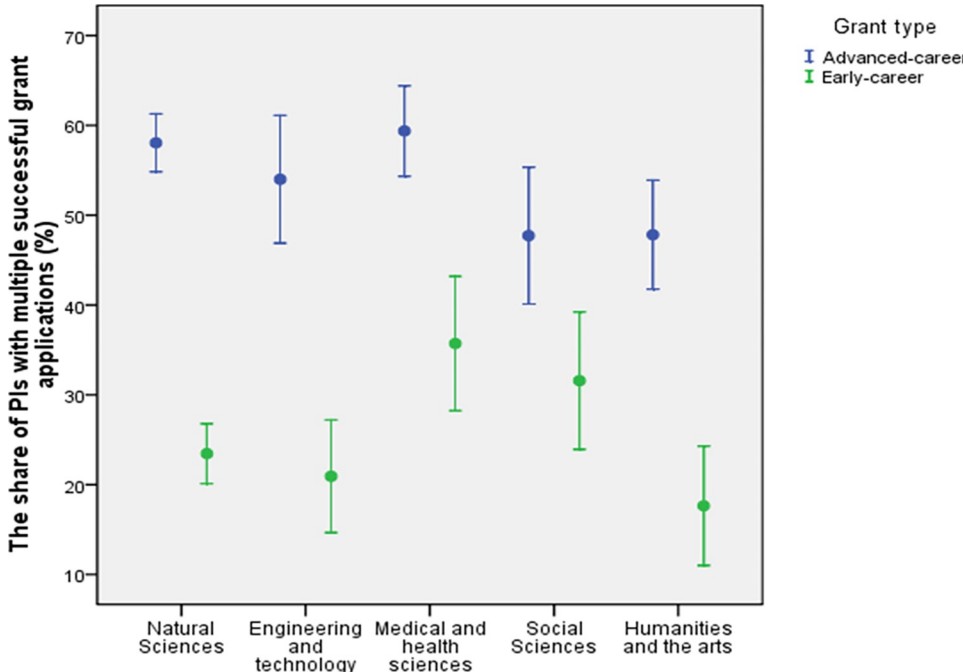

**Fig 11. The share (± SE) of PIs, who have gained the grant several times across grant type and Frascati fields through all grants awarded by the Estonian Research Council between 2013–2022.** Agricultural and veterinary sciences are not shown because of low number of early-career grant holders with several grants (2 PIs).

Across all science fields, the average age at the time of obtaining the PhD was smaller for those PIs, who have gained the grant several times (Fig 12). This holds both for early and advanced-career levels ($P < 0.05$ for both comparisons, two-sample Wilcoxon rank-sum tests). This result is logical, as the application for grant is based on having a PhD–the earlier one gains it, the more she or he can apply for grants. That said, linear and fast progression in academic system has its advantages.

## Discussion

We have generated a unique dataset of all national grant PIs between 2013–2022, based on personalized information present in the Estonian Research Information system (www.etis.ee). Well developed, openly and digitally accessible national grant data of Estonian research allows generating a holistic picture of research funding within a country, which is often not accessible to larger countries. That said, Estonian case brings relevant learning points also to other countries, both in terms of methodological approaches and policy-making. We are able to signpost four main learning points, first three tightly connected to RQ1-RQ3, fourth one reports universal key takeaway from the descriptive statistics on annual R&D budgets and success rates.

### Age: Erosion of the concept of "early-career" in funding decisions and increased funding of retired academics (RQ1)

Average age of the PI varied significantly across disciplines (Table 3, Fig 6). It is striking to see the large variation in age, 26 to 61 yr. for the early-career grant category (Table 3). In addition, the age of advanced-career level grant holders also varied strongly, from 29 to 76 yr. (Table 3, and Fig 9). These results obviously redefine what is early-career (one may gain early grant and soon retire according to the Estonian retirement age—64 yr.). Albeit it is a sensitive aspect in

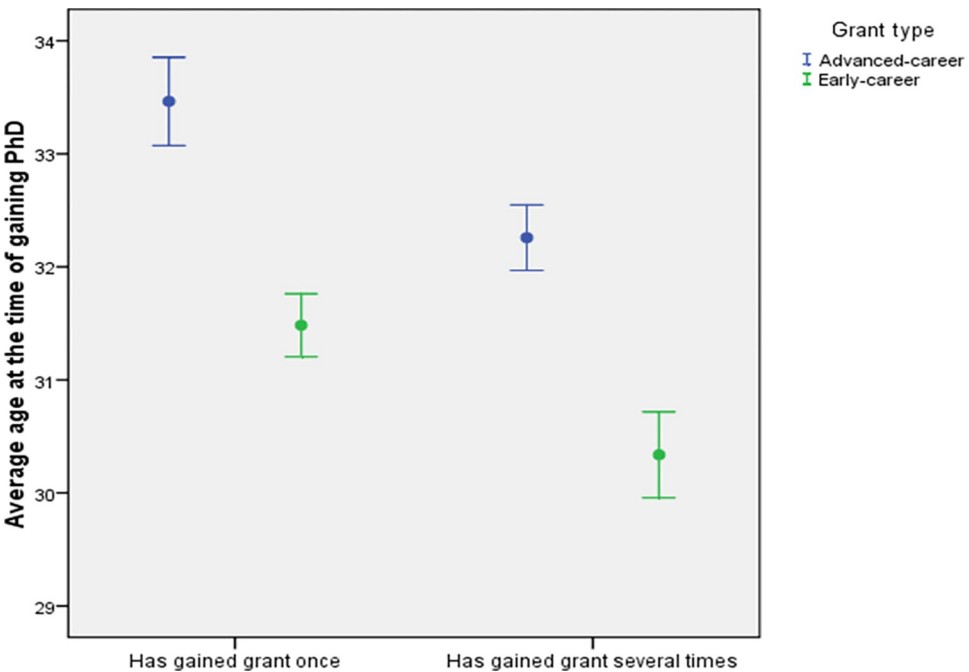

**Fig 12. Average age at the time of gaining PhD (± SE) in relation to success of obtaining the grant once or multiple times for different grant types (early vs. advanced).** The comparisons are based for all grants awarded by the Estonian Research Council between 2013–2022.

grant allocations, it is an important one–roughly 20% of academic workforce in Estonia is 60 or older (35). Young early-career academics may fall out of competition if they are competing with older academics who have been within the system for decades. Factually, participating in the scientific system and gaining scientific capital and defending the PhD at the late life stage gives an unfair edge of competition in a system where the only criterion of eligibility for early-stage grant is the time after PhD.

We see it to be not the problem of one characteristic (the age), but a challenge that grounds on "intersectionality" phenomena. Age in our case correlates with long work experience in the academic system, thus often also having a good research track. That said, the age parameter interconnects with research productivity and so called insider advantage as compared to the young early-career researchers who have been with the system only for let's say 4–6 years (as compared to 30 years). A reasonable solution here would at the early-career grant level to consider only those publications that have been published during or after PhD. At the moment, Starting Grants are evaluated according to the following rule: "the research activities of the applicant during the past 10 years" [22]. As very young applicants usually start their publication track during the PhD studies or even after the studies, they are faced with unequal competition when they are applying next to older academics who have solid track of publications past 10 years (these academics just defended their PhD at later stage of life, yet publication track started much earlier). We would recommend taking an example from ERC Starting Grants, where it states how PIs may present up to five publications in their application.

## Research excellence: Weak seniority effect (RQ1+RQ2)

Although newer studies have revealed how high research impact may happen at every age and every stage in academic career [29], still the seniority is often seen as a sign of greater research

productivity [13]. That said, we could also expect how PIs of advanced grant outperform early-career PIs, yet this was not always the case. Our study revealed how the seniority effect was rather weak—the average ages at the time of the grant differ significantly between early- and advanced-career levels, but the research productivity does not necessarily grow by age (Fig 8). Connected to the previous point–a great share of PI-s report rather low research productivity (see Figs 8 and 9, especially in Social Sciences, Table 4). This is strongly at odds with previous stock of literature, according to which, research productivity ought to grow with age [13].

Our data shows (see Table 3) how at the early-career level, the average age of the PI was the highest in Social sciences (38.5 years) and the lowest in Natural sciences (34.3 years). Previous studies may explain the high age of PIs in case of junior level grants in Social sciences with the fact that on average, a doctorate in Social sciences in Estonia is received by the age of 39, while in Natural sciences it is 34 [30, 31]. This is also in line with larger share of studies that confirm how especially Humanities and the arts, but also Social sciences reveal remarkably higher age of doctorates than doctorates from other fields [32]. This in turn sets an interesting challenge to policy making as it questions universality of the early-career level grant allocations, where depending on the discipline, the early-career grant PIs from one discipline may on average be at the age when people from other disciplines would already apply for an advanced grant. Merely by the age, a junior PI in Social sciences may well be a senior for Natural sciences. Field specific differences may also be explained by variations in career trajectories–people from Social sciences and other practice oriented fields often not only enroll to doctoral studies at later stages of life [31, 33, 34], but also make more career brakes, moving between industry and academia [33], or even working only part-time for the academia.

## Fluctuating scientific productivity (RQ2)

Grant writing with very low success rates "impose a substantial opportunity cost on researchers by wasting a large fraction of the available research time for at least half of our scientists, reducing national scientific output, and driving many capable scientists away from productive and potentially valuable lines of research" [6]. The small differences in the score in the end depend on the degree of subjectivity of the reviews [14, 35, 36]. We noticed how to some extent, the fluctuations in research excellence among funded grants (especially at the advanced grant level) seem to overlap with the change in evaluation bodies. For example, there was sharp decrease in research excellence between 2013–2017 and in 2022, period that overlaps with the nomination of the new Expert Panel and Evaluation Committees, who are final decision-makers on who gets funded (see Fig 8).

Although in this paper we were not able to compare successful applicants with non-successful ones, there is past evidence to suggest how grant success does not automatically lead to better research excellence. To illustrate the point, a study form van den Besselaar and Sandström [37] based on early-career grant analysis to clearly bring out how "the grant decisions seem to have no predictive validity, as the successful applicants have no higher performance than the best performing non-successful applicants". They do show how "the granted applicants have a much better academic career", yet "the growing evidence that predictive validity of panel and peer review of grant applications is difficult to establish".

## The RPRT effect may be fatal to early-career researchers

We agree with Murray et al. [38] in claiming how "one must be clear on the source of the problem that may often be rooted in limited national funds for research". Applying at the right time is crucial as the amount of research money allocated each year depends on how many research grants started in the previous years and in the current year. In short, how much

money has been freed up for new projects–this is clearly seen in how greatly the annual budget for new grants differs by years, variations may be 7-fold (see Fig 3). If a person applies for the grant at the time when a lot of money has been "occupied", the success rates are much lower than in a year when many grants have ended, "money has been freed". That said, we can see how the random element in grant success is largely determined by the fact how one needs to be in the right place at the right time, i.e. independently of research performance. The probability of being funded is greater in years when there is more funding available and more projects will be funded as convincingly demonstrated in our analysis (Fig 3).

What further makes the situation highly unjust is that the time-period for application for early-career level grants is very limited. For example, in the case of the starting grant, one can only apply two to seven years after getting the PhD degree, and if this "window" overlaps with overall modest annual allocation of funding in Estonia, RPRT effect has the strongest consequences to early-career researchers. Overall the peculiarity of Estonian grant system is the fact how early-career grant system has limited time window, yet advanced grant system is open far beyond the retirement age, close to the age of 80 (oldest grant recipient was 76).

What we see from the Estonian example is that national funding bodies should make clear what is the aim of these grants. As well expressed by Bendiscioli [39],

> "The question whether peer review is good at selecting the best people or proposals is linked to the goals of the funding scheme. Is the aim to identify future success, to reward past success or to achieve specific scientific or social goals?"

Every national funding system may have its own unique setting, and the aims of funding should align with it. Estonian researchers have low state funding and high dependence on research money, where being successful in attracting grants may literally determine your next months salary. If the national grant system is a battle for survival, then what should be the aims of the funding bodies? To fund only those who are most likely to succeed with their topic? To fund those, who already have a good research track record and have gained grants previously (this seems to be the current logic)? To fund younger applicants as to lessen the ageing of the academic labour market? Etc.

## Conclusions

The share between competitive and non-competitive research funding largely varies among countries. In countries with high share of competitive funding, there is a risk that the research funding becomes excessively concentrated. This can curb the entry of early-career researchers into the research system and also have adverse effects on the diversity of the research landscape.

The strongest takeaway from the study is a need to redefine what is the end purpose of the national grant system. Over the years, early-career, startup grants have been given to researchers with age varying from 26 to 61 yr., reflecting the erosion of "early" in research career, especially given that in several cases, the allocation of startup grants to preretirement stage researchers was associated with a recent second academic degree. In the case of advanced-career level grants, the grant-holders age at the time of awarding the grant has varied from 29 to 76 yr. with the trend of increased share of retired grant holders. Thus, coupled with highly fluctuating and modest national R&D funding, these grants weakly serve the needs of entry of young researchers, tenure and promotion of academic workforce. The analysis suggests that great annual fluctuations and abnormalities in grant allocations have far-reaching consequences for individual career prospects, especially for the younger generation of researchers, and might also question the sustainability of the overall national academic labor market.

Our research raised important questions for future–faced with limited amount of national funding on research, what should be the optimal strategy in terms of heterogeneity in how many would gain funding and how much. This optimal point is unique for each national setting, depending on the whole picture. We agree with previous studies [6, 8, 9, 11, 17, 40] in claiming how there is a need for research addressing how scarce research funds are allocated. As so far, the greatest share of studies addressing this questions have been conducted based on large and English based national systems like Canada [38, 41, 42], Australia [10, 12], UK [43, 44], and U.S. [6, 11, 14, 45], or capture the recipients of elite grants like ERC [13, 46, 47], and the most addressed disciplines have been medical and STEM fields [8, 43, 44]. The scholarly community would benefit greatly by additions from smaller and non-English countries like Estonia and similar, and capturing also less studied research disciplines.

Our study revealed how in case of very small and ageing academic labour market, countries need to pay special attention to the unwanted side-effects of grant systems, e.g. facilitating grant funding of near to retirement age academics (both at the early and advanced level!), possibly on the expense of new academic generations. This abnormality, coupled with evaluation bodies that comprise of only people from the same small academic system–greatly harnesses the transparency of the grant allocations.

Lastly, our study has also limitations. First, due to un-access to individual scores we were not able to see how the characteristics of the PI and the review scores would correlate. Second, it would be valuable to compare successful and non-successful applicants and especially, do their research excellence differ after (not) gaining the grant. We might expect how those high level applicants who did not gain the grant have even stronger need to work twice as hard to apply for the next ones–thus, research excellence might rise, compared to the funded applicants. Again, due to limited access to data we were not able to test this hypothesis.

## Supporting information

**S1 Table.**
(TIF)

**S1 Dataset. Listing anonymised grant dataset.**
(XLSX)

**S1 Fig. Boxplot of the age at the time of the grant by grant type (early vs. advanced) and Frascati fields across all grants awarded by the Estonian Research Council between 2013–2022.** 1 –Natural sciences; 2 –Engineering and technology; 3 –Medical and health sciences; 4 –Agricultural and veterinary sciences; 5 –Social sciences; 6 –Humanities and the arts.
(TIF)

**S2 Fig. The distribution of the original data of the number of 1.1 articles across grant types (early/advanced) for all grants awarded by the Estonian Research Council between 2013–2022.** Smaller graphs show the distribution after logarithmic transformation.
(TIF)

**S3 Fig. Boxplot of the normalized number of articles by grant type (early vs. advanced) and Frascati fields across all grants awarded by the Estonian Research Council between 2013–2022.** The outliers with the normalized number of articles above 10 are removed to construct the graph. 1 –Natural sciences; 2 –Engineering and technology; 3 –Medical and health sciences; 4 –Agricultural and veterinary sciences; 5 –Social sciences; 6 –Humanities and the arts.
(TIF)

## Author Contributions

**Conceptualization:** Eneli Kindsiko, Kärt Rõigas, Ülo Niinemets.

**Data curation:** Eneli Kindsiko, Kärt Rõigas, Ülo Niinemets.

**Formal analysis:** Eneli Kindsiko, Kärt Rõigas.

**Methodology:** Eneli Kindsiko, Kärt Rõigas, Ülo Niinemets.

**Resources:** Eneli Kindsiko.

**Software:** Kärt Rõigas.

**Supervision:** Ülo Niinemets.

**Visualization:** Eneli Kindsiko.

**Writing – original draft:** Eneli Kindsiko, Kärt Rõigas.

**Writing – review & editing:** Eneli Kindsiko, Kärt Rõigas, Ülo Niinemets.

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
