## [Decision Letter · Decision Letter 0]

10 Jan 2022

PONE-D-21-17096Hypercompetition for research grants: To the best of the best or just for the ones who are lucky?PLOS ONE

Dear Dr. Kindsiko,

Thank you for submitting your manuscript to PLOS ONE. After careful consideration, we feel that it has merit but does not fully meet PLOS ONE’s publication criteria as it currently stands. Therefore, we invite you to submit a revised version of the manuscript that addresses the points raised during the review process.

We look forward to receiving your revised manuscript.

Kind regards,

Gemma Elizabeth Derrick, Ph.D

Academic Editor

PLOS ONE

Journal Requirements:

2. Please consider changing the title so as to meet our title format requirement (https://journals.plos.org/plosone/s/submission-guidelines). In particular, the title should be "Specific, descriptive, concise, and comprehensible to readers outside the field" and in this case it is not informative and specific about your study's scope and methodology.

Reviewers' comments:

Reviewer's Responses to Questions

**Comments to the Author**

1. Is the manuscript technically sound, and do the data support the conclusions?

Reviewer #1: No

Reviewer #2: No

2. Has the statistical analysis been performed appropriately and rigorously? 

Reviewer #1: No

Reviewer #2: N/A

3. Have the authors made all data underlying the findings in their manuscript fully available?

Reviewer #1: No

Reviewer #2: No

4. Is the manuscript presented in an intelligible fashion and written in standard English?

Reviewer #1: Yes

Reviewer #2: No

5. Review Comments to the Author

Reviewer #1: This article has one strong advantage: it is based on a comprehensive proposal database (all competitive research proposals submitted for funding in Estonia over a nine year period) and corresponding bibliometric data on those submitting these proposals. The statistical analysis of these data is extremely well done (with one very minor concern about using full publication counts instead of fractional counts because of high multi-authorship in the large national physics lab in Estonia). The data provides excellent information about proposal success rates by field. These data are also presented in the broader context of policy changes in Estonia. This is an extremely solid foundation on which to build upon.

The biggest weakness in the article is that the major claims of the authors (vis-à-vis hyper-competition and luck) are not supported by the data. Following is a clarification of these two concepts; why this reviewer doesn’t see support for the author’s claims about hyper-competition and luck and corresponding implications for going forward.

Hyper-competition:

The concept of hyper-competition was introduced into the field of management in 2010 by Richard D’Aveni. This concept has a very specific meaning. Basically, it describes a situation where competitive advantages are less sustainable (the rich do not automatically get richer). Hyper-competition is also associated with changes in the broader competitive environment (more rivalry among firms- less long term relationships with suppliers and customers) and detectable changes in the temporal persistence of profitability (a more direct indicator that competitive advantages are less sustainable). An excellent discussion of the concept and supporting literature on hyper-competition can be found on Wikipedia.

The authors use of the word ‘hyper-competition’ is rooted in two articles that are outside the field of management: Folcher [2016] and Edwards [2017]. Both articles use the word ‘hyper-competition’ to raise a concern about how competitive advantage in academia has changed. Folcher noted how the current practice of using quantitative (usually bibliometric) indicators for faculty promotion and tenure are “replacing deeper considerations of the quality and novelty of work, as well as substantive care for the societal implications of research”. Edwards is concerned that these quantitative methods are resulting in perverse behavior that, when added to decreased funding rates, will result in unethical behavior by researchers and the de-legitimization of academic research. Their use of the concept of ‘hyper competitive’ is slightly different from D’Aveni. Edwards & Folcher are more concerned with a shift in what determines competitive advantage (from qualitative to quantitative indicators) and the corresponding effect on research outputs (less novelty, less concern for societal needs, more unethical behavior).

These are two different uses for the same word. Quantitative indicators for the promotion of academics might actually result in competitive advantages that are more sustainable over time- not less sustainable. Edwards & Folcher are actually describing conditions where the basis for success is changing- they are not using the original definition of the concept. These are important distinctions to make going forward.

Luck vs. Excellence (in getting a proposal funded):

The authors are concerned about two interrelated issues: the ability to identify excellent proposals and the corresponding effect of funding decisions on the survival of early stage researchers.

The ability to evaluate the excellence of a proposal is where luck (vs. excellence) comes into play. One has to look at the distribution of proposal evaluations (by expert and by field) to make this claim. I’m not aware of any publications on this- but anecdotally (which means I’ve had the opportunity to see unpublished data on this question), the distribution is very non-linear. There’s usually a very small group of proposals where all the evaluators consider the proposal as ‘excellent’. There’s also a very small group where there is consensus that the proposal is not. But the vast majority of proposals are in the middle group- there isn’t a consensus that this is an excellent or poor proposal. Luck doesn’t play a role in the two extreme cases: funding decisions are based on expert consensus. Luck plays an important role for the vast majority of proposals where evaluators don’t agree. And it is in the group of ambiguous proposals that secondary criteria (such as gender equity, diversity of funded institutions, diversity in the benefits of research and career stage) can be applied.

Survival deals with whether a researcher in Estonia symbolically lives (gets a 5 year grant so they can continue to do research for another 5 years) or symbolically dies (they don’t get funding after trying two or three times- they have to shift to a different career path). To put this in concrete terms, the authors mention that there were about 3000 researchers and 3800 one-year grants in Estonia (I’m basing this on the reported 766 grants with an average 5 year time period). So over this nine year period, there’s only enough funding to support each researcher for 15 months. This leads to a system that could range anywhere from systemic economic starvation (everybody is given one grant that can support them for 15 months- and this has to last 9 years) to extreme economic and social stratification (a few hundred fully funded researchers; the rest have nothing unless they work for the fully funded researchers). Survival of those just starting their career is of particular concern. It’s not just about the age of the work force. It’s also about whether economic and social power is becoming more concentrated. It's about Folcher's point that novelty (and new ideas) might be driven out of the system.

Given these two issues- the authors’ logic seems to be the following. If all proposals across all fields are treated equally (the best proposal gets funding), then one might claim that funding is based on excellence. There will still be deaths- but the survivors represent ‘the best of the best’. But as one goes deeper and deeper into the non-consensus group of proposals, funding decisions are more based on luck. In concrete terms, you are especially lucky if you happen to have been an engineering early career researcher in 2018- when the Estonian government decided to invest more money in research that supports job-creation in the manufacturing sector. As the authors put it- you were in the right place at the right time.

Weaknesses of this study:

My concern is that the data provided by the authors do not support their central premise of hyper-competition and luck.

Lower proposal success rates are not necessarily related to hyper-competition. Building on D’Aveni, there’s no evidence that faculty shifted their efforts from forming alliances with funders to attacking their peers (this is one indicator of hyper-competition) and there’s no evidence that the current way to maintain competitive advantage (i.e. publish more in the peer review articles) is deteriorating. Building on Folcher, there’s no evidence that grant decisions were only based on the quantitative rankings of the reviewers (the authors provided no analysis of the quantitative ranking of the proposals and the role of quantitative vs. qualitative evaluations in making the final funding decision). Building on Folcher, there’s no evidence that researchers are less novel after proposal success rates dropped. Building on Edwards, there no evidence of increased unethical behavior. In other words- the data used in this study doesn’t support the fundamental claim or concerns that the environment is hyper-competitive.

Nor does the data used by the authors support the central premise of luck. Given the reasonable assumption that, in any field, there is a very small percentage of excellent proposal, a very small percentage of extreme poor proposals and a vast middle ground of non-consensus proposals, low proposal acceptance rates will always result in a higher percentage of excellent proposals. Nor does low proposal acceptance rates automatically result in reduced diversity if there is a sufficient pool of non-consensus proposals by which to implement diversity policies.

Recommendations:

Moving forward, the authors need to be more precise about what they mean by ‘hyper-competition’ and ‘luck’ and make a far stronger link between their claims and the analyses they provide.

I suggest that the authors incorporate proposal evaluation scores in their analysis. This is critical if one is going to make claims about luck. I doubt that the final decisions on proposals, where there was expert consensus, significantly differed from the quantitative evaluation. The emphasis needs to be on those proposals where evaluators didn’t agree. This is where choices are closer to being random- and actual choices may be biased in a variety of ways (gender, institutional reputation, economic vs. societal benefit, etc.).

More importantly, the authors have an opportunity to provide fundamental insights into the survival strategies of early career researchers. This would require that they shift to the use of survival analysis (which is well supported by STATA- the statistical package they are using). They probably have all of the data that is needed to do this. One possibility is to look at the different survival strategies that early career researchers are using (such as being an individual researcher that tends to work alone and requires very little funding vs. being employed in a biochemistry lab or a massive physics lab that has extremely expensive equipment that needs to be maintained). They would need to define what is really meant by ‘death’. Is it the abandonment of a career path in research? Is it the failure to publish a research paper over the next few years? We know very little about the survival strategies of those that received insufficient funding. What really happens to them? The authors of this paper have the opportunity to provide unique insights into the factors that increase (or decrease) the likelihood that an early career researcher will symbolically die.

In summary, I cannot recommend that this article be published as currently structured. The database and analysis, while excellent, is disconnected from the central claims and concerns about hyper-competition and luck. There’s no evidence that reduced proposal acceptance rates results in hyper-competition using the definition by D’Aveni (a deterioration of how competitive advantage is established), Folcher (reduction in novelty) or Edwards (increased gaming of the system). There’s no evidence that more proposals are funded based on luck (this would require an analysis of the proposal evaluation scores by each evaluator and in each field). There are logical reasons to assume that lower acceptance rates will result in funded proposals that are more likely to be excellent. Diversity does not automatically decline when proposal acceptance rates are lower- it might persist and actually increase if secondary criteria are applied to the non-consensus proposals.

Reviewer #2: # Overview

## What are the main claims made in the paper?

The authors claim that hypercompetitive research environments make funding allocations random and are therefore detrimental to research excellence and propose to examine this claim in Estonia. To support their claim, the authors collated data from public databases listing details of the awards and of the awardees (via http://www.etag.se) and PI demographic information (via http://www.eetis.ee).

## Is the topic important?

The issue of whether research grant funding allocations are promoting research excellence is very topical and highly significant, especially in the tumultuous economic times we are currently experiencing. To examine this in a Northern European Country such as Estonia, is also unusual and could add a diversity of data and perspectives, which the field of research policy needs.

# Major issues

## Contextualisation

A first major issue is the fact that the claims were not well contextualised. There is no review of what hyper-competition in grant funding allocations means, how it has been identified in the past literature, and how it manifests itself in the current research policy landscape: is it the same in all regions of the world? The authors seem to imply that low success rates are the evidence for hyper-competition but this is a little too simplistic as a definition (low success rates could result from an increase in the volume of applications, an increase in the quality threshold, an increase in budgetary constraints, and so forth).

The second part of the claim is that hyper competition turn funding allocations in a lottery (by making them "random"). Again, it is not clear what past evidence can support this claim. It may "feel" this way to those who are not successful but this does not mean that the funding decisions allocations are random. For example, the authors claim that decisions are made based on very small differences in scores, but do not review research which can warrant this claim.

The last part of the claim is that hyper competition and random allocations of funding are detrimental to research excellence. Here again, there is no clear contextualisation of that research excellence means, and whether there is past evidence to show that the standard approach to fund research (e.g., via individual peer-reviewers informing panels and the grant funders) is actually detrimental to innovation and contribution to knowledge.

## Data and claims

A second major issue is the fact that the data did not support the claims. Instead the data presented showed various descriptive facts about the grant funding allocations and a change in policy which had an impact on the type of applications received. By and large, the data provided a plausible picture: early career researchers get a grant when they are younger, with less papers although the actual numbers of papers vary with disciplines. Prior success in getting a grant is a strong predictor of future success, although this also varies across disciplines.

None of these data, however, directly provide evidence for the claims advanced. There is evidence that success rates fluctuates but this could also be due to the economic context rather than a deliberate attempt to increase competition. Not only that, success rate seems to have increased steadily since 2018. The evidence of a trend in increasing hyper-competitiveness is not compelling (especially since there are many possible causes for the fluctuations in success rates). Similarly there is no data to evidence the claim that allocations are random or that they are not supporting research excellence (however this may be defined). The reader is left asking themselves: "So what?" on many occasions.

# Other issues

## Discussion

A lot of the claims in the discussion seem unrelated to the evidence in the results section. For example "Our study confirmed the importance of luck in hypercompetitive research fund allocation" (lines 428-429). I do not see how the authors could infer this from the data they provided. One might as well say that the study "confirmed the importance of the excellence of the PI in hypercompetitive research fund allocation." Neither is implausible, neither can be substantiated by the data available. This is just an example. The same applies to all the claims in the discussion.

## English writing

The manuscript needs careful proofreading as it shows several instances of grammatical errors or sentences which would need rewriting to improve readability.

## Open data

The authors argue that the database they put together cannot be shared because of GDPR concerns. But I do not see why the authors could not share the anonymised data which underpin their graphs, especially if the data was collated from public sources.

6. PLOS authors have the option to publish the peer review history of their article (what does this mean?). If published, this will include your full peer review and any attached files.

Reviewer #1: No

Reviewer #2: No

---

## [Author Response · Author response to Decision Letter 0]

21 Feb 2022

We submitted a separate response letter to reviewers.

---

## [Editor Report · Decision Letter 1]

6 Oct 2022

PONE-D-21-17096R1Grant funding of early-career researchers in a highly fluctuating environment: shifting from excellence to luck and timingPLOS ONE

Dear Dr. Kindsiko,

Thank you for submitting your manuscript to PLOS ONE. After careful consideration, we feel that it has merit but does not fully meet PLOS ONE’s publication criteria as it currently stands. Therefore, we invite you to submit a revised version of the manuscript that addresses the points raised during the review process.

 The points to be addressed can be seen as minor in work required, but extremely important conceptually.  Specifically, even though you have done consideration work in defining your conceptualisation of hyper-competition in the Estonian context, there are still problems with how your claims in the discussion are sufficiently supported by the results show. In particular, I still find how you equate 'life experience' with 'academic experience' as underpinning your claims that how a swing towards success in grant allocation to older academics (arguably close to retirement) discriminations against young early-career researchers.   Here, the use of the work 'discriminates' is problematic in that it assumes that older PhD graduates have an unfair advantage over younger ECRs.  Perhaps a more nuanced interpretation is that the funding agencies do not distinguish between older and younger ECRs based on age.  This would focus your argument more appropriately on problematising the purpose of ECR funding, and the definition to an ECR, rather than assuming that there is a source of discrimination.  Your findings support the former conclusion more robustly than the later. Adjusting the argument in the Abstract, introduction and conclusion would address this issue sufficiently. I see this point as one of argument and for further debate, however this debate must extend from the paper.  I would hate for any debate (of which the PLoS platform allows) to be distracted by the above issue, rather than contributing to a more robust problematisation of the function of the Estonian competitive grant and research system. 

We look forward to receiving your revised manuscript.

Kind regards,

Gemma Elizabeth Derrick, Ph.D

Section Editor

PLOS ONE
---

## [Author Response · Author response to Decision Letter 1]

24 Oct 2022

EDITOR: The points to be addressed can be seen as minor in work required, but extremely important conceptually. Specifically, even though you have done consideration work in defining your conceptualisation of hyper-competition in the Estonian context, there are still problems with how your claims in the discussion are sufficiently supported by the results show.

REPLY: We highly appreciate the feedback. In a revised version, we have given the above mentioned points remarkably more attention in the paper. 

EDITOR: In particular, I still find how you equate 'life experience' with 'academic experience' as underpinning your claims that how a swing towards success in grant allocation to older academics (arguably close to retirement) discriminations against young early-career researchers. Here, the use of the work 'discriminates' is problematic in that it assumes that older PhD graduates have an unfair advantage over younger ECRs. 

REPLY: We acknowledge the value-laden essence of the word “discrimination”. In the paper it is mentioned only in one place, under discussion:

“If grant allocation prefer older academics who have good record of articles, yet defended their doctoral degree around 50-60 years of age, this discriminates young early-career academics, who do not have such an advantage.” In a new version we have removed it. 

To comment the claim “Here, the use of the work 'discriminates' is problematic in that it assumes that older PhD graduates have an unfair advantage over younger ECRs”, it is not so much the age, but the fact how older ECRs do have more publications, because they are often the academic staff who have been working in university without a degree, but doing research in a similar manner as other. 

EDITOR: Perhaps a more nuanced interpretation is that the funding agencies do not distinguish between older and younger ECRs based on age. This would focus your argument more appropriately on problematising the purpose of ECR funding, and the definition to an ECR, rather than assuming that there is a source of discrimination. Your findings support the former conclusion more robustly than the later.

REPLY: We agree 100%. It is more based on established research track than merely on age. Yet, we see it to be not the problem of one characteristic, but a challenge that grounds on “intersectionality” phenomena. Age in our case correlates with long work experience in the academic system, thus having also good research track. That said, the age parameter interconnects with research productivity and so called insider advantage as compared to the young early career researchers who have been with the system only for let`s say 4-6 years ( as compared to 30 years). A person with 6 years experience and track in publishing is competing with a person with 30 years of experience. 

We have now taken the focus away from the discrimination, but through the intersectionality phenomena we address the point how early career grants fail to fulfil their core mission – helping those who are at the very beginning at their career. In fact, the main difference between the early career and advanced career grant is the expected experience within the academic system and the publication track. People with 30 year experience are more eligible to advanced level competition with their research track, yet they apply for early career track, because there they can overshadow the young ERCs who haven’t got such an advantage. 

EDITOR: Adjusting the argument in the Abstract, introduction and conclusion would address this issue sufficiently.

REPLY: In the revised version we have adjusted the argument not only in abstract, introduction and conclusion, but also in the discussion. 

EDITOR: I see this point as one of argument and for further debate, however this debate must extend from the paper. I would hate for any debate (of which the PLoS platform allows) to be distracted by the above issue, rather than contributing to a more robust problematisation of the function of the Estonian competitive grant and research system.

REPLY: We strongly agree with the point. We see it as a question of our perhaps poor wording – as mentioned above, it is not a discrimination about the age, but a wider, intersectionality challenge. People with decades of experience and long list of publications are competing in the same pool with people who are literally new to the system. It undermines the mission of the early grant system, especially in case of a national research system that has very limited funding. 

Best regards,

The authors

---

## [Editor Report · Decision Letter 2]

26 Oct 2022

Getting funded in a highly fluctuating environment: shifting from excellence to luck and timing

PONE-D-21-17096R2

Dear Dr. Kindsiko,

We’re pleased to inform you that your manuscript has been judged scientifically suitable for publication and will be formally accepted for publication once it meets all outstanding technical requirements.

Kind regards,

Gemma Elizabeth Derrick, Ph.D

Section Editor

PLOS ONE